# A Visual and VAE Based Hierarchical Indoor Localization Method

**DOI:** 10.3390/s21103406

**Published:** 2021-05-13

**Authors:** Jie Jiang, Yin Zou, Lidong Chen, Yujie Fang

**Affiliations:** College of Systems Engineering, National University of Defense Technology, Changsha 410073, China; jiejiang@nudt.edu.cn (J.J.); zouing14@163.com (Y.Z.); 17719498319@163.com (Y.F.)

**Keywords:** indoor localization, computer vision (CV), variational autoencoder (VAE)

## Abstract

Precise localization and pose estimation in indoor environments are commonly employed in a wide range of applications, including robotics, augmented reality, and navigation and positioning services. Such applications can be solved via visual-based localization using a pre-built 3D model. The increase in searching space associated with large scenes can be overcome by retrieving images in advance and subsequently estimating the pose. The majority of current deep learning-based image retrieval methods require labeled data, which increase data annotation costs and complicate the acquisition of data. In this paper, we propose an unsupervised hierarchical indoor localization framework that integrates an unsupervised network variational autoencoder (VAE) with a visual-based Structure-from-Motion (SfM) approach in order to extract global and local features. During the localization process, global features are applied for the image retrieval at the level of the scene map in order to obtain candidate images, and are subsequently used to estimate the pose from 2D-3D matches between query and candidate images. RGB images only are used as the input of the proposed localization system, which is both convenient and challenging. Experimental results reveal that the proposed method can localize images within 0.16 m and 4° in the 7-Scenes data sets and 32.8% within 5 m and 20° in the Baidu data set. Furthermore, our proposed method achieves a higher precision compared to advanced methods.

## 1. Introduction

Indoor localization has recently been the focus of much attention due to its wide commercial application value, providing services such as navigation and positioning [1], search and rescue [2], advertising pushing, and location socialization in large and complex indoor environments (e.g., museums, airports, and supermarkets). Unlike outdoor positioning, the use of GPS in indoor environments is generally highly limited due to signal occlusion, thus making accurate positioning a complicated task. Therefore, researchers employ indoor signal transceiving devices such as Bluetooth beacons [3], Wireless Fidelity (Wi-Fi) [4], Digital Enhance Cordless Telephone (DECT) [5], and Radio Frequency Identification (RFID) [6]; however, these external devices are required to be placed in the environment in advance, resulting in additional installation and maintenance costs. If the external device of the system is altered, the entire system must also be updated. Visual-based localization is an alternative localization method that can be positioned using just a pre-built model and a single camera without any other external devices required. Vision-based localization can be applied to 3 Degree of Freedom (3DoF) positioning services, as well as additional applications such as intelligent robots [7] and virtual reality [8] that require high-precision 6DoF pose estimation. This can typically be accomplished by direct image-based localization via a 3D model based on sparse feature points from simultaneous localization and mapping (SLAM) [9,10] or Structure from Motion (SfM) [11].

Direct image-based localization initially extracts the local features (e.g., edge, corner, and line) of the database images and calculates their 3D positions. This is followed by the matching of the 2D local features of the query image with the 3D points in the model, also known as 2D-3D matching, which is frequently achieved by using a nearest neighbor search in the descriptor space. Such algorithms perform well for small-scale scenes, yet for larger scales such as airports, the descriptor space in the database increases, and the matching process requires an excessive amount of computing resources. This presents difficulties for platforms with limited resources. Moreover, indoor environments contain a large number of repetitive appearances and elements such as corridors, doors, and windows, and are thus prone to mismatches, resulting in poor pose estimation results.

In order to overcome the aforementioned limitations, numerous methods [12,13] have been proposed with a particular focus on reducing the amount of 2D-3D matching. These include making use of global features such as color, texture, and shape. For example, the two-stage localization approach first employs an image retrieval procedure to determine candidate images in the database similar to the query image and subsequently performs 2D-3D matching to calculate the 6DoF camera pose. Such a method is efficient due to the reduced space required for the nearest neighbor search. The development of convolutional neural networks (CNNs) has greatly improved the performance of image retrieval procedures, yet these systems still cannot directly estimate the camera pose with centimeter accuracy. The majority of current image retrieval tasks [14,15] often use supervised or weakly supervised networks for training. However, this results in difficulties in obtaining data labels (e.g., GPS information) for indoor environments, with manual labeling increasing labor costs. Therefore, based on its convenience and simple implementation, we employed an unsupervised training network variational autoencoder for the prior image retrieval in our system.

In particular, in the current paper, our hierarchical localization framework learned global and powerful local features for visual-based localization in order to achieve large-scale indoor and accurate 6DoF pose estimation tasks. Similar to humans inferring their position in the natural environment, we initially applied the overall appearance characteristics to infer the approximate location, and subsequently determined where we were by using some local and notable visual cues. Therefore, we used hierarchical localization [16] (Figure 1) for large-scale indoor scenes, where we used global features to achieve the rough localization and local features to achieve accurate pose estimation. The key contributions of this paper are summarized as follows:We designed a hierarchical localization framework that utilizes an unsupervised network VAE and SfM, which only requires RGB images as the input.We demonstrated that our proposed method can achieve 6DoF pose estimation and has higher localization accuracy than most deep learned methods.

The rest of the paper is organized as follows. We first briefly review the related literature in Section 2. Section 3 presents the framework of the proposed unsupervised indoor localization system. Section 4 presents the specific methods, including using the SfM pipeline to build 3D models, constructing a VAE model and the corresponding optimization scheme for training, calculating the pose, and selecting the best pose. Section 5 presents the experimental results, which show that our method stands out. Finally, we present a discussion and conclude the paper in Section 6.

## 2. Related Work

### 2.1. Visual-Based Localization

Visual-based localization methods can generally be categorized into regression-based, structure-based, and image retrieval-based methods. Regression-based methods include end-to-end visual localization models trained by deep learning that are able to directly obtain the regressed 6DoF camera pose [17,18,19,20]. However, such methods are not applicable for the visual localization of large-scale scenes and are associated with low accuracies [21]. Although some methods [22,23] have made improvements and the accuracy has been greatly improved, these kinds of method all need to use camera pose data.

Structure-based methods use feature matching between query image and map to obtain the 6DoF pose. The 3D map is mostly constructed by SfM methods [11,24,25]. The pose of image query is computed by matching key points in the query image and 3D points in the 3D map, and then solving the Perspective-n-Points (PnP) problem. In this type of method, in addition to traditional pose calculating methods, most methods also need to use labeled data for training. For example, DSAC (differentiable RANSAC (Random Sample Consensus)) [26] and DSAC++ [27] need RGB-D data, and BTBRF (Backtracking Regression Forests for Accurate Camera Relocalization) [28] also needs camera pose for training. However, the searching and matching calculations increase with the points increase in the 3D map. In order to improve the efficiency of such approaches, researchers have proposed several solutions including vocabulary trees [29], prioritized searches [30], and remote server calculations [31]. However, the benefits of these methods are limited and they are not suitable for resource-constrained mobile platforms and large-scale scenarios. In addition, local features lack the ability to capture the global context of the image and require the robust aggregation of points to effectively achieve pose estimation.

Image retrieval-based methods use either global features or visual vocabulary to match query and database images, and subsequently obtain the approximate position of the image from visually similar images. The implementation of such methods is carried out based on global descriptors, traditionally defined as aggregated artificial features such as Scale Invariant Feature Transform (SIFT) [32], Bag-of-Visual-Words (BoVW) [33], and vector of locally aggregated descriptors (VLAD) [34,35]. Although these methods only need RGB images, they have complex computation and are not competitive with deep learning methods [36,37]. With the development of deep learning, the learned global features DEep Local Features (DELF) [14] and NetVLAD [38] have greatly improved the performance of image retrieval tasks, principally due to the establishment of large labeled visual data sets. The retrieved position can be directly used to calculate the pose between the query and the retrieved image, yet the discretization of the database means that the pose can only be approximated.

Moreover, the retrieved images can be further used to restrict the search space of large maps for the structure-based method [39,40]. The fusion of hierarchical localization [41] and knowledge distillation [42] can successfully realize large-scale localization on platforms with limited computing resources. InLoc [12] is the latest image retrieval scheme based on dense information, using depth features to first retrieve the most similar images and subsequently estimate the 6DoF camera pose. The aforementioned methods combine image retrieval-based and structure-based methods in order to overcome these shortcomings, determining the approximate position of the query image instead of the precise 6DOF pose. However, these methods require labeled data for training, such as camera pose, which increase the cost of annotation, and some methods need special data such as depth image, which increases the difficulty of data collection.

### 2.2. Variational Autoencoder

A variational autoencoder [43] is an unsupervised deep generative model that embeds high-dimensional information such as images or text into low-dimensional latent variables through encoder and decoder networks.

Denote the model input as x and the latent variable as z. A VAE learns stochastic mappings between an observed x-space and a latent z-space, where the input distribution is typically complicated while the latent variable distribution can be relatively simple. The encoder network is used to encode an image to latent variable z, where the stochastic encoder is denoted as the inference model, $q_{\varphi }(z|x)$, with parameters φ. The decoder network aids in the decoding of latent variable z to an image similar to the input image. The decoder is an integral intractable posterior and denoted as $p_{\theta }(x|z)$ with parameters θ. The generative model on x can be determined by marginalizing out latent variable z as follows:(1)$$p_{\theta }(x)=\int p(z)p_{\theta }(x|z)dz.$$

Such an implicit distribution over x can be quite flexible. If z is discrete and $p_{\theta }(x|z)$ is a Gaussian distribution, then $p_{\theta }$ is a mixture-of-Gaussians distribution [44]. Similar to other variational methods, the optimization object of the variational autoencoder is the evidence lower bound (ELBO), also known as the variational lower bound. The log-likelihood ELBO of the data set is calculated using the approximate posterior and the Kullback–Leibler (KL) divergence between the true and approximate posterior:(2)$$\log p_{\theta }(x)=\underset{ELBO}{\underset{︸}{L_{\theta ,\varphi }(x)}}+\underset{KL-divergence}{\underset{︸}{D_{KL}\left(q_{\varphi }(z|x)\;||\;p_{\theta }(z|x)\right)}}$$
where the second term in Equation (2) is the KL divergence and is non-negative. Thus, the ELBO is a lower bound of the log-likelihood of the data set:(3)$$\begin{array}{ll} \log p_{\theta }(x) & \geq L_{\theta ,\varphi }(x)\\ & =\log p_{\theta }(x)-D_{KL}\left(q_{\varphi }(z|x)\;||\;p_{\theta }(z|x)\right) \end{array}$$

Therefore, the VAE can be trained by maximizing the ELBO. The first term on the right-hand side of Equation (3) can be considered as a reconstruction term that aims to maximize the expected data log-likelihood $p_{\theta }(x|z)$ and the posterior estimate $q_{\varphi }(z|x)$. The second KL divergence determines the gap between the ELBO and the likelihood $\log p_{\theta }(x)$, and the smaller the gap, the better the approximation of the true (posterior) distribution $p_{\theta }(z|x)$ by $q_{\varphi }(z|x)$. KL divergence can be treated as a regularization term in network training and can prevent $q_{\varphi }(z|x)$ from collapsing to a single point.

Unlike autoencoders (AEs), VAE does not encode the training data as an isolated vector, rather it can force the latent variable to fill the space [45]. Therefore, the input images can be encoded in latent variables via the encoder network, which is useful for image retrieval [46] and clustering [47,48] tasks. The semantic visual localization [49] encodes the semantic 3D voxel volumes into latent variables and chooses latent code μ as global descriptor for city-level visual localization.

## 3. System Overview

Our proposed unsupervised hierarchical indoor localization method only requires RGB images. The localization process is similar to the way in which humans determine their position in the natural environment. More specifically, we initially apply the overall appearance characteristics to infer the approximate location and subsequently determine where we are by using some local and notable visual cues. Figure 2 presents the workflow of our localization system, which consists of two key processes: the preprocessing and the localization.

For the preprocessing, we use the database images to train the VAE network, and the trained network is then applied to generate global features of the database images and query image for subsequent image retrieval tasks. We simultaneously extract the local features from the database images and use SfM to reconstruct a sparse 3D model of the scene. The local features and the 3D model are crucial for the pose estimation. For the localization process, given a query image, we first generate the global features through the trained VAE network and subsequently perform a global retrieval to determine similar images in the database. These images are denoted as candidate images and are clustered into different clusters by covisibility clustering [41]. We then perform 2D-3D local feature matching between the key points of the query image and 3D points of each cluster in order to calculate the camera pose through these matches. The 6DoF pose that comes from the cluster with the most inliers is selected as the best pose.

## 4. Methods

### 4.1. Preprocessing

Offline preprocessing includes image collection, 3D modeling, VAE designing and training. At this stage, we collect the images from the scene to form the image database. Next we use the images to reconstruct the 3D model, which is the key point during the localization process. At the same time, we will design a VAE network and use a reasonable training scheme to train.

#### 4.1.1. 3D Modeling

Accurate pose calculation and localization tasks require a pre-built 3D map. Compared to other types of data such as depth and infrared data, an RGB image is easy to obtain, even using a simple smartphone. Thus, we only use RGB images to achieve 3D modeling. We employ the open source COLMAP [11,24], currently the most widely adopted incremental SfM scheme, and use unordered and ordered images to build a sparse point cloud model. COLMAP employs siftGPU (the graphics processing unit (GPU) version of SIFT) as the local feature. Moreover, the local features of each image and all 3D points can be stored in a database for their efficient management.

#### 4.1.2. VAE Structure Design

Figure 3 depicts the VAE structure design. The convolutional VAE network takes an image as the input and encodes it in latent variable z, then it will be decoded back to an image. The encoder network contains four convolutional layers with 3 × 3 kernel and 2 × 2 stride to achieve downsampling. Note that maxpooling and other deterministic spatial functions are not performed here. A hyperbolic tangent activation layer is followed by each convolutional layer. At the end of the encoder network, two fully connected layers are used to output mean μ and standard deviation σ, which represent the posterior distributions of the latent variables. The re-parameterization trick is then adopted to generate samples from μ and σ, where ϵ∼N(0,I). These samples are pushed into the decoder network as inputs, where the decoder network maintains the same kernel size and number of strides, while the convolutional layers are replaced by deconvolutional layers.

#### 4.1.3. Training and Optimization

Our model aims to learn global latent representations of images. We can optimize the variational lower bound objective in Equation (3) to achieve global feature learning for our system. The bound can be grouped into two terms: the reconstruction term and the KL divergence term:(4)$$L_{recon}=-\log p_{\theta }(x),$$
(5)$$L_{KL}=D_{KL}\left(q_{\varphi }(z|x)||p_{\theta }(z|x)\right),$$
(6)$$L_{loss}=L_{recon}+L_{KL}.$$

The VAE model can be trained by optimizing $L_{loss}$ in Equation (6) via the gradient descent method. A fine trained model can encode useful information from images into latent variable z and will have a non-zero KL divergence term and a relatively small reconstruction term. However, the straightforward training of VAE can suffer from posterior collapse [45,50], and it fails to make use of enough information. When the posterior collapse phenomenon occurs, the model ends up relying solely on the auto-regressive properties of the decoder while ignoring the latent variables, which become uninformative. This means that the latent variables will no longer represent the features of input image and the latent space will be inaccurate for each latent variable. This phenomenon often occurs when encoding high-dimensional information such as images and texts into the latent space by using the VAE. This occurs when the variational distribution closely matches the prior for a subset of latent variables, that is, the variational distribution collapses toward the prior. It can be represented as $\exists i\;s.t.\;\forall x{\;q}_{\varphi }(z_{i}|x)\approx p(z_{i})$. The phenomenon can occur during training that the KL divergence vanishes and the cost function value tends to zero. This means that the latent variables will no longer represent the feature of input image, and the latent space will be inaccurate for each latent variable. Thus, the distance between latent variables cannot be used to measure differences between input images.

Therefore, we adopt KL annealing [51] and free bits [52] to overcome posterior collapse. During the training process, the KL divergence term is multiplied by weight β, which ranges from 0 to 1. When the training begins, β is set to zero, such that the network can learn more information from the input images and encode this into the latent variables. The weight is then linearly increased at a certain training step until it reaches 1, where the cost function becomes the original VAE cost function. The free bits method is a modification of ELBO, with a minimum information constraint applied to the latent variable. This ensures that each latent variable dimension can keep a minimum number of bits of information and allows for a greater amount of information to be encoded. The latent dimensions are divided into K groups and M minibatches. Equation (7) describes the modified objective, demonstrating that using fewer than λ nats of information per subset j is not advantageous.
(7)$$L_{KL}=\beta *\sum_{j=1}^{K}\max\left(\lambda ,E_{x∼M}\left[D_{KL}\left(q_{\varphi }(z_{j}|x)||p_{\theta }(z_{j}|x)\right)\right]\right).$$
where β∈[0,1] during the training processing. K is generally set to equal its individual dimension, and for all j there exists $E_{x∼M}\left[D_{KL}\left(q_{\varphi }(z_{j}|x)||p_{\theta }(z_{j}|x)\right)\right]\geq \lambda$, with the KL annealing effectively preventing the posterior collapse.

### 4.2. Localization

#### 4.2.1. Prior Image Retrieval

We employ the VAE to perform the prior image retrieval task as it does not require labels and data preparation is minimal. The VAE encodes the images into the latent space and the mean μ of latent variable is then used as the descriptor of the global features for the prior image retrievals. After the VAE network is trained, it can be used as a global feature extractor to perform unified feature extraction on the images stored in the database to build a feature database. When calculating pose, the VAE network is used to extract the global features of the query image, and the extracted global features are used as the judgment basis to perform similarity matching with the image features stored in the database. When performing image retrieval tasks, the feature database has been established, we use NN (nearest neighbors) search to achieve it.

#### 4.2.2. Pose Estimation

After prior image retrieval, we obtain the candidate images through image retrieval achieved by the VAE network. We use the local features of these candidate images to calculate the precise 6DoF pose. The construction of the 3D model and the subsequent pose estimation are based on siftGPU. However, despite its strong performance in pose calculation, its computational costs are high, particularly in large-scale indoor environments, where the increased 3D model points may result in mismatching and long computation. Thus, in order to speed up the localization and increase its accuracy, we restrict the search space of the local 2D-3D matches by reducing the number of 3D points considered. Prior image retrieval can achieve this, but the number of 3D points of candidate images is still excessive. As indoor environments exhibit similar and repetitive features, the retrieved candidate images may belong to different locations having the same global features. Therefore, we cluster the candidate images into different clusters according to covisibility clustering in order to reduce the number of 3D points. As shown in Figure 4, cameras that can see the same 3D points are clustered into the same cluster [13]. The 3D points of each cluster are the sum of the points of their images (camera).

We extract the 2D key points of the query image and match them with the 3D points included in each cluster. The 2D-3D matching is a PnP problem that is solved via RANSAC. This outputs a robust pose estimation and an evaluation of the geometric consistency of the resulting 2D-3D matches. The RANSAC scheme outputs a robust pose estimation and an inlier number. The inlier number is used for the evaluation of the geometric consistency of the resulting 2D-3D matches. If the inlier number is less than the set threshold, it is considered as a failed pose estimation and the localization is failed. Now we can get the camera pose and inlier number through each cluster, after all clusters are iterated, the one with the largest inlier number is selected as the optimal cluster, and the corresponding camera pose is the optimal 6DoF pose of the query image.

## 5. Implementations and Evaluation

In this section we evaluate our proposed method on the 7-Scenes data set and Baidu localization data set.

### 5.1. Implementations

We describe the experiments performed using the two data sets to evaluate the proposed localization system. We aim to prove the high localization accuracy and efficiency of our scheme for large-scale indoor environments. The parameter setting in different stages of different data sets is shown in Table 1.

(1) Data sets

The 7-Scenes data set [53] is a collection of tracked RGB-D camera frames in seven small indoor scenes, where each sequence set is split into distinct training and testing sets. The images were captured using a Kinect RGB-D camera with a 640 × 480 resolution. The spatial size of the 7-Scenes data set is small, the images are all from small indoor spaces, and the model size is smaller than about 4 cubic meters. Here, we only utilized the RGB images for 3D model building and global descriptor extraction. As the images were camera frames, consecutive images were highly similar and had a short baseline, resulting in high computational costs and a complicated initialization. Therefore, we selected an image from every five frames as the training set.

The Baidu Institute of Deep Learning (IDL) indoor localization data set [54] contains images captured in a Chinese mall. It occupies over 5000 square meters and the length is around 240 m. These images are challenging for visual localization due to objects in motion, repetitive scenes, and reflective structures. The data set contains 689 RGB images captured from a Digital Single Lens Reflex (DSLR) camera as the training set and over 2000 query images captured from different types of cell phones as the testing set. We used the provided images and corresponding camera, while the LIDAR data were omitted from our work. The training images had a fixed 2992 × 2000 resolution, which was distinct from those of the testing images, such as 2064 × 1161, 1632 × 1224, 2104 × 1560 and 2104 × 1184.

Note the input data from the two data sets were images and camera poses from the training and testing sets. In order to evaluate our methods, we used the known poses reconstruction method in COLMAP to establish the models in our experiment. The training data and testing data in the data set were registered in the same world coordinate system $O_{1}$ at the time of collection. Using the conventional image reconstruction method in COLMAP, the training data were registered in a different world coordinate system$\;O_{2}$. However, the testing data were still registered in the $O_{1}$ coordinate system and could not be used as ground-truths. In the localization application, there was no need to evaluate the performance of the method, and the conventional image reconstruction method could be used, which only required RGB images. The camera poses from the testing set were employed as the ground-truths for the results.

(2) Data preprocessing

Prior to the training of the model, we performed several preprocessing steps on the input images. The input size of the network described in Section 4.1.2 was 64 × 64 × 3, thus we downsized the 7-Scenes images to the same size. The Baidu images were downsized to 128 × 96 × 3, cropped into four corners and a center with a size of 64 × 64 × 3. This cropping approach increased the amount of data for network training and did not influence the global features. Before being fed into the network, the images were normalized to range from 0 to 1 in order to prevent an ill-conditioned model and to facilitate convergence.

(3) Training

We implemented the proposed model using TensorFlow and trained the network for 80 k (k refers to 1000 and 80 k is 80000) iterations with a batch size of 50. Following 20 k iterations, we performed the KL annealing mentioned in Section 4.1.2, and at 40 k the weight β reached 1. According to paper [52], common values of λ include [0.125, 0.25, 0.5, 1, 2], and here we set λ=1, which allowed for the model to preserve enough information. We adopted Adaptive Moment Estimation (Adam) [55] to optimize and set the learning rate as 1 × 10^−4^, with other parameters based on recommended settings. The model was implemented on a desktop with Intel i7-7700k CPU, 16G RAM, and 6 GB GPU memory with NVIDIA GTX1060.

(4) Metrics

We evaluated the localization accuracy by comparing the estimated and reference poses that were derived from the ground-truth of the data set. We used the positional (m) and angular (°) parameters to represents the differences. We also compared the run time in seconds with different localization methods.

(5) Methods

During the prior image retrieval step, the trained VAE model generated 128-dimensional descriptor vectors, while for the Baidu data set this value was 640. We used the nearest neighbors search to retrieve 50 or 100 database images as candidate images. In the covisibility clustering step, we clustered the candidate images into different clusters according to the image covisibility relationship in the COLMAP database. In the pose estimation step, we employed the P3P-RANSAC implementation and set the reprojection error to 10 pixels. The one having the largest number of inliers was selected as the optimal cluster, and the corresponding camera pose was the optimal 6DoF pose of the query image.

### 5.2. Evaluation Results

#### 5.2.1. 7-Scenes Data Set

The 7-Scenes data set consisted of seven small indoor scenes. In order to verify the localization performance of our method for large scenes, we assumed that these scenes came from the same building. However, as the scenes had distinct world coordinate systems, we established seven 3D models respectively rather than integrate them into a global model. Then we put all training images together to train the VAE network, thus it could extract all global features. Taking the Stairs data set as an example, given a query image, first we retrieved the top 50 images from all training images and only selected those images belonging to the Stairs training set as the candidate images (because we established seven models, it was convenient to carry out covisibility clustering and visualize retrieval results). We then clustered these candidate images, calculated the camera pose, and compared the result with the ground-truth data.

Figure 5 depicts the behavior of the KL divergence term during the 7-Scenes training with KL annealing. The loss exhibited an early drop in training corresponding to the cheap encoding of information into latent variables by the model, followed by a marked rise as the full KL divergence penalty was paid, and the subsequent gradual reduction as more information was encoded into latent variables.

VAE exhibited a good data clustering performance, with the higher dimension features extracted by its convolutional layers reduced to a lower dimension. The images from the same cluster were generally mapped to the same area of latent space. Figure 6 depicts the t-SNE plot of the latent variables determined from 7-Scenes. The categories exhibited clear boundaries, and similar images, particularly the sequence frames, were clustered together. This indicated the effective extraction of low-dimensional features by the model, allowing the latent variables to be used for the image retrieval. In order to show the image retrieval effect intuitively, we made a cumulative proportion plot for each separate testing set in Figure 7. For example, as for the Stairs testing set, taking one of the test images as the query image, in the prior image retrieval stage, the top 50 images were retrieved from the 7-Scenes training set. The number of images belonging to the Stairs data set (candidate images) was recorded. We localized each image in the Stairs testing set and recorded the corresponding number of retrieved candidate images; the cumulative proportion is plotted in the blue line in Figure 7. Other data sets are plotted in the same way, and concave lines have better retrieval results than convex lines.

Figure 8 is a visualization display of the localization results on 7-Scenes. In order to display it conveniently, we visualized the localization results every five images.

In addition to using the methods mentioned above, we also added a comparative experiment that used the basic VAE (no optimized training scheme). Then we compared our approaches to the deep learned methods, with the localization results reported in Table 2. The reported results of baselines of other methods were directly transferred from their papers. Following the reasonable image retrieval and covisibility clustering, the search space for local feature matching was reduced, and P3P-RANSAC presented a good pose estimation result. Our method exhibited a high accuracy with centimeter-level position errors and single-digit angle errors. Furthermore, it had lower position error and orientation error and performed better than most advanced deep learned methods, and could localize images within an average of 0.16 m and 4°. The results showed that without the optimized training scheme, the basic VAE applied in our pipeline showed lower localization accuracy compared to the optimized training VAE.

#### 5.2.2. Baidu Data Set

The Baidu data set contained realistic, large-scale and challenging indoor scenes. There were few training images (689 images), the training and testing images came from different cameras with varying viewpoints, and environments included scenes with light changes and reflective structures. Deep state-of-the-art approaches performed poorly and it was difficult to improve localization results using deep learning methods based on Objects-of-Interest (OOIs) [56], and it successfully localized 23% of the images at 1 m and 5°, and 40% within 5 m and 20°. However, OOIs required manual annotated planes. Structure-based COLMAP was able to localize more images, with approximately 45% at 1 m and 5°, and 58% at 5 m and 20°. Figure 9 compares our method with the current advanced methods. PoseNet [17] and DSAC++ [27] could not localize enough images due to a lack of training data. Our method successfully located 17.4% images at 1 m and 5°, and 24% at 5 m and 20° when retrieving 50 candidate images. When the retrieved images were increased to 100, the localization success rates increased to 22.6% and 32.8%, respectively, as shown in Ours (100). If we did not use the mentioned training scheme, our proposed methods with basic VAE had lower localization success rates.

Figure 10 presents the image retrieval results from the Baidu data set. Once similar images were retrieved, our method successfully localized the query images with a high accuracy comparable to current advanced supervised depth regression methods. Our methods performed slightly worse than the structure-based method as the majority of images were not successfully retrieved. Unlike the 7-Scenes data set, the Baidu data set did not contain frame sequences and had fewer training images. This consequently resulted in a poor retrieval performance. Although COLMAP exhibited a higher localization accuracy, it was not competitive in terms of running time, with our proposed method running faster (Table 3). Moreover, our method was associated with a rapid global search time, which also proved its feasibility in localization tasks for large-scale environments.

## 6. Conclusions

In the current paper, we proposed a novel unsupervised hierarchical localization method that integrated learned global features with handcrafted local features. The system had a good accuracy for indoor scenes and performed better in large-scale scenes, and was convenient and easy to implement. The key contribution of this paper is the unsupervised network VAE and traditional pose calculation that only require RGB images as the input. This localization scheme provided a new approach for the advancement of indoor localization. Since the localization accuracy of our method depends on the performance of the image retrieval, a lack of training images limited the performance of our method. Constructing a new VAE model with a good retrieval accuracy will be the focus of future research. In addition, artificial features are still the main basis for pose calculation. They occupy a larger memory and computation will also consume a lot of time. Therefore, the problem of deployment on mobile platforms was not completely solved, so in the next step we will consider using deep learned local features or line features that occupy less memory to further optimize the algorithm.

## Figures and Tables

**Figure 1 sensors-21-03406-f001:**
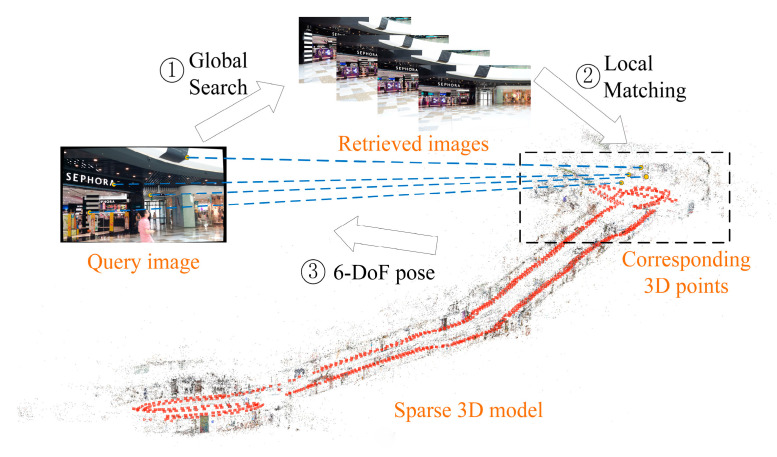
Hierarchical localization. Candidate images are retrieved from the database, and then the 2D key points of the query image are matched with the corresponding 3D points of the candidate images in model.

**Figure 2 sensors-21-03406-f002:**
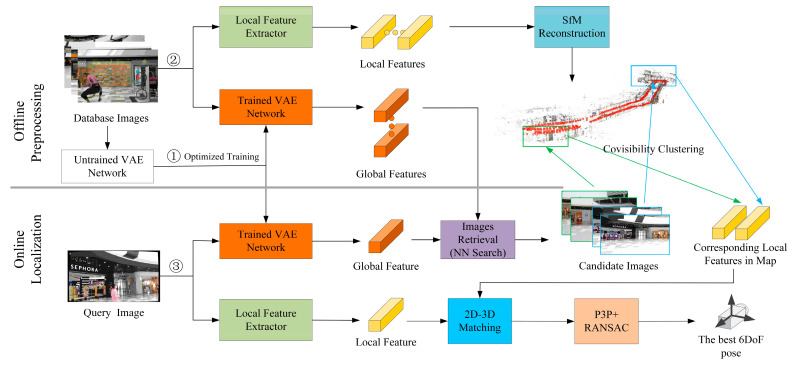
Overview of the unsupervised hierarchical indoor localization system. The preprocessing includes step 1 and step 2, and the localization process is step 3.

**Figure 3 sensors-21-03406-f003:**
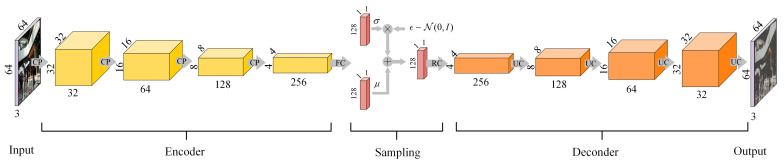
Structure of the designed variational autoencoder. CP represents Convolution + Pooling, FC represents Fully Connected, RC represents Reshape + Convolution, UC represents Upsampling + Convolution. The mean μ of latent variable forms our global descriptor.

**Figure 4 sensors-21-03406-f004:**
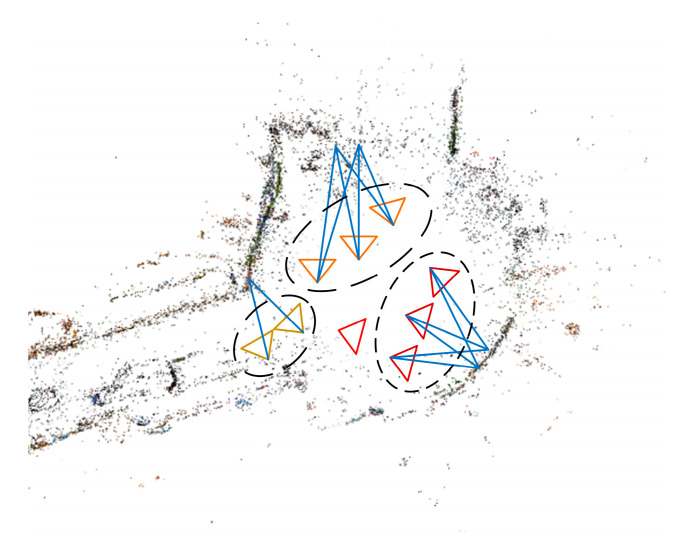
Covisibility clustering. Cameras are clustered into three clusters marked by different colors, where the red is the camera of the query image.

**Figure 5 sensors-21-03406-f005:**
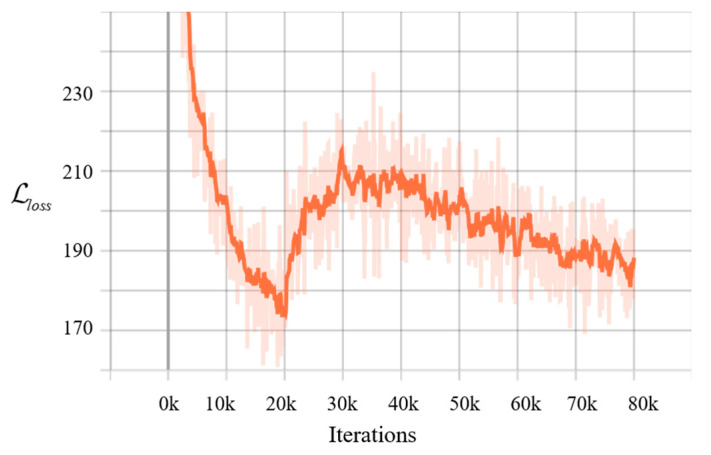
Training loss with KL annealing. Weight β gradually increases from 0 to 1 between 20 k to 80 k iterations.

**Figure 6 sensors-21-03406-f006:**
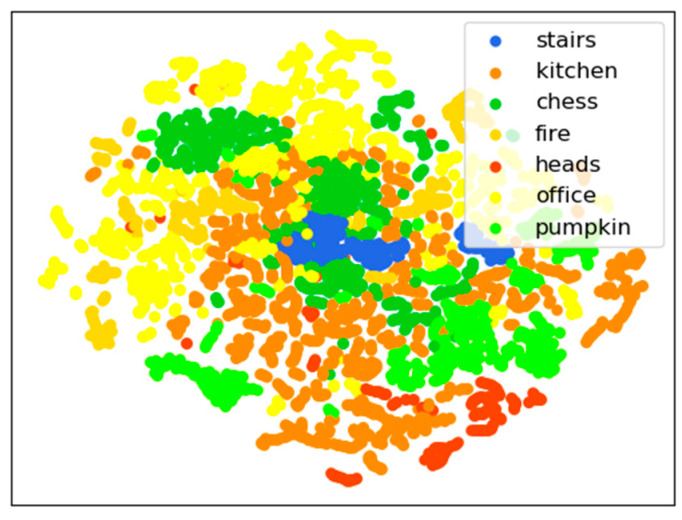
Latent space of the 7-Scenes dimensionality reduction based on t-SNE [56].

**Figure 7 sensors-21-03406-f007:**
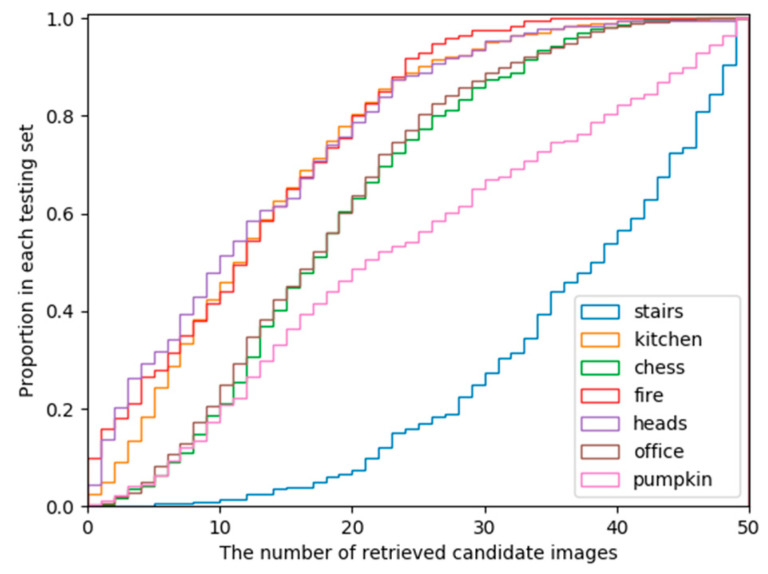
Cumulative proportion plot for each separate testing set (top 50 images are retrieved).

**Figure 8 sensors-21-03406-f008:**
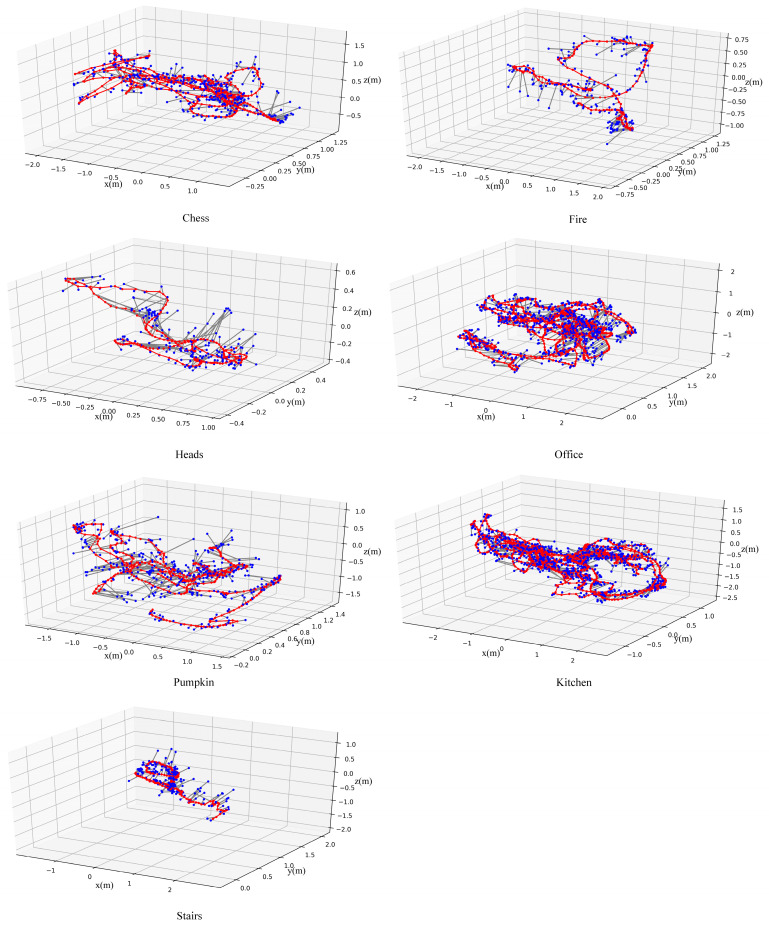
Localization for testing (blue) and training (red) data from the 7-Scenes data set. The red line is the trajectory of the ground-truth of the test image, and the black line represents the difference between the calculated by result and the ground-truth.

**Figure 9 sensors-21-03406-f009:**
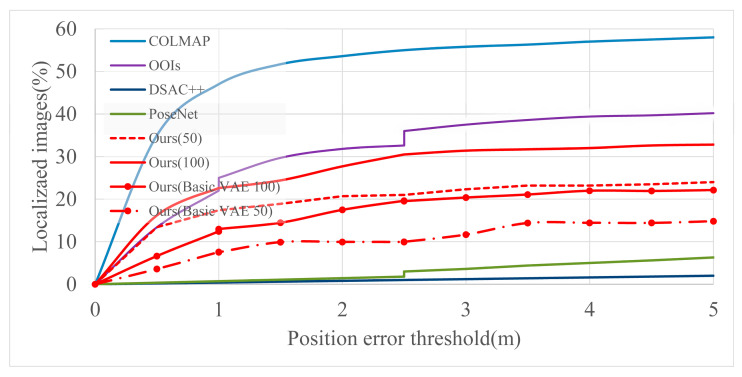
Percentages of localized images from the Baidu data set across position error thresholds. The reported results partly come from paper [56]. The orientation error threshold is 5°, 10°, and 20° for position errors below 1 m, between 1 m and 2.5 m, and above 2.5 m, respectively. Ours (50) means we used the optimized training scheme and retrieved 50 images in prior image retrieval stage. Ours (100) means we used the optimized training scheme and retrieved 100 images in prior image retrieval stage. Ours (Basic VAE, 50) means we did not use the optimized training scheme in the VAE training stage and retrieved 50 images. Ours (Basic VAE, 100) means we did not use the optimized training scheme in the VAE training stage and retrieved 100 images.

**Figure 10 sensors-21-03406-f010:**
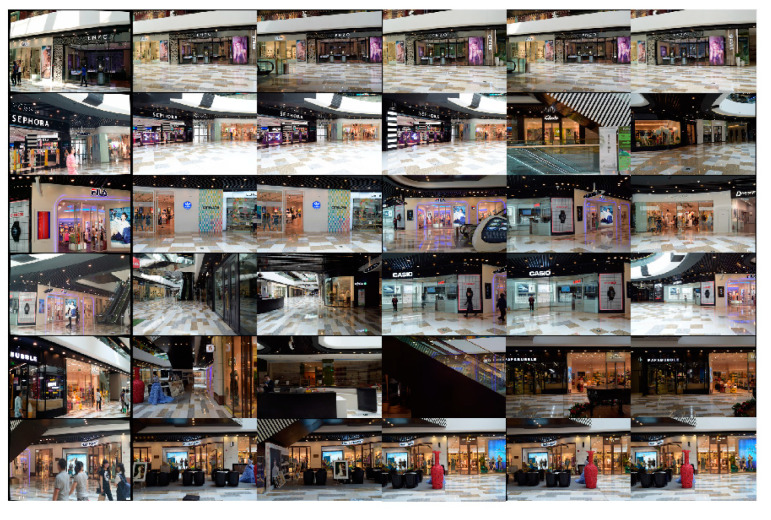
Image retrieval results from the Baidu data set. The first column includes the query images, followed by the top five retrieved images.

**Table 1 sensors-21-03406-t001:** Parameter settings in different stages of different data sets.

	Data Sets	7-Scenes Data Set	Baidu Data Set
Parameters			
Training images resolution (pixel)		640 × 480	2992 × 2000
Testing images resolution (pixel)		640 × 480	2064 × 1161, 1632 × 1224 2104 × 1560, 2104 × 1184
Downsize resolution (pixel)		64 × 64 × 3	128 × 96 × 3
Input size of VAE (pixel)		64 × 64 × 3	Cropped into four corners and a center with a size of 64 × 64 × 3 (5 images)
Dimension of global descriptor (dimension)		128	640
Batch size		50	50
learning rate		1 × 10^−4^	1 × 10^−4^
λ		1	1
KL annealing beginning Moment (iteration)		20 k	20 k
Number of retrieved images		50	50 (or 100)
P3P-RANSAC reprojection error (pixel)		10	10

**Table 2 sensors-21-03406-t002:** Comparison of accuracy under the 7-Scenes data set using the mean position/orientation error (m/°). Ours (Basic VAE, 50) means we did not use an optimized training scheme in the VAE training stage and retrieved 50 images. Ours (50) means we used the optimized training scheme and retrieved 50 images in the prior image retrieval stage.

	Methods	Relative PN [18]	RelocNet [19]	Dense-VLAD [35]	Mobile-PoseNet [20]	Improved CNN-Based Pose Estimation [22]	Image-Similarity-Based Method [23]	Ours (Basic VAE, 50)	Ours (50)
Data Sets									
Chess		0.13/6.46	0.12/4.14	0.21/12.5	0.17/6.78	0.17/5.34	0.21/5.73	0.42/7.25	0.12/2.32
Fire		0.26/12.7	0.26/10.4	0.33/13.8	0.36/13.0	0.30/10.36	0.40/12.11	0.55/8.72	0.15/3.07
Heads		0.14/12.3	0.14/10.5	0.15/14.9	0.19/15.3	0.15/11.73	0.25/14.38	0.44/10.26	0.11/3.64
Offices		0.21/7.35	0.18/5.32	0.28/11.2	0.26/8.50	0.27/7.10	0.30/7.58	0.53/8.57	0.16/2.54
Pumpkin		0.24/6.35	0.26/4.17	0.31/11.3	0.31/7.53	0.23/5.83	0.37/7.46	0.59/9.38	0.16/2.33
Kitchen		0.24/8.03	0.23/5.08	0.30/12.3	0.33/7.72	0.29/ 6.95	0.42/7.11	0.55/8.65	0.14/2.37
Stairs		0.27/11.8	0.28/7.53	0.25/15.8	0.41/13.6	0.30/8.30	0.36/11.82	0.62/12.53	0.16/2.34

**Table 3 sensors-21-03406-t003:** Comparison of running time in second. Ours (50) means retrieval of 50 images in prior image retrieval stage, and Ours (100) means retrieval of 100. Record of the running time of the entire localization of COLMAP and running time in different stages of our methods.

	Processes	Global Search	Covisibility Clustering	Feature Match	P3P-RANSAC	Total
Methods						
Ours (50)		0.12	0.07	0.98	1.53	2.70
Ours (Basic VAE, 50)		0.12	0.07	0.98	1.53	2.70
Ours (100)		0.24	0.14	2.26	1.49	4.13
Ours (Basic VAE, 100)		0.24	0.14	2.26	1.49	4.13
COLMAP						47.61

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
