# Peer review of "A Visual and VAE Based Hierarchical Indoor Localization Method"

_sensors, 2021, doi:10.3390/s21103406_

Round 1
Reviewer 1 Report
The paper proposes a hierarchical method for improving the localization in largescale indoor scenarios. The work is quite exhaustive and well presented. However, some minor changes are suggested:
1) Revise Section 4.2. Some incorrect terms for describing the 3D orientation and reconstruction procedure and methods are present (e.g., "disordered and ordered images", or "COLMAP model").
2) Expand the Conclusion section, discuss further limitations of your method, and how you plan to solve these issues (i.e., what is planned in the future researches for improving the procedure).
Reviewer 2 Report
The manuscript deals with a visual-based localization framework that exploits global and local features according to VAE (Variational AutoEncoder) and SfM (Structure from Motion) methods.
The topics are interesting but the manuscript is confused and unclear. In addition, some language errors make the manuscript difficult to read.
The authors fail to clearly describe what is the main problem discussed in the manuscript and how to solve it. Some preliminary considerations about the problem are missing while a lot of important references are never cited (see below).
The proposed framework includes a lot of different algorithms but their description and exploitation is very confused and unclear. I suggest the authors to make use of flowcharts, diagrams and pseudo-code fragments to describe the proposed methods and their interactions, and to summarize the parameters adopted in the evaluation section.
The evaluation section is very confused. Please clarify how the proposed framework compares against the other ones and if the comparison is carried out using the same image database or not.
In summary, the manuscript looks like a technical report, not a scientific paper. I suggest the authors to rewrite it completely by addressing the issues described below.
Specific issues
lines: 46-48, 70-72, 75-78, 82-83, 95-97, 157-159, 171-174, 182-184, 194-197, 256-262, 267-276, 329-336, 379-382, 421-427 : unclear/obscure parts;
lines: 55-61, 63-64, 89-92, 200, 215-216, 246, 334, 367-374, 385: missing references;
lines 66-68: other unsupervised methods are available. Please discuss your choice;
lines 84-85: very generic. Please specify and clarify;
line 87: the paper organization is missing;
eq. (2): the meaning of q(.) is unclear;
eq. (3): it is trivially derived from (2). It is not clear why it is inserted here. Please clarify;
line 162-163: is this method rubust w.r.t. changes in the scenario?
line 165-166: pre-processing and localization steps are not clearly identified in figure 2;
line 179: what is "covisibility clustering"? Is there any missing reference here?
figure 3: RC? FC? These terms are never defined;
eq. (4)-(6): not clear;
line 219: check the inline equation;
line 229: which is the original VAE equation?
line 307-319: not clear. Use flowchart and summarize algorithm parameters in a table.
line 357: what is t-SNE?
table 2: it not clear.
Reviewer 3 Report
The paper presents an hierarchical system of video-based localization, in which a VAE is used as a feature extractor. The paper is written in a pleasent way for the reader, it adresses a relevant problem in a way that seems promising and appears to have an interesting performance trade-off between localization accuracy and computetional effort needed. The authors have used public datasets and report their results having as baselines multiple other existing methods, which is a very strong asset of this work. Nevertheless, there are points were significant improvement is needed, which could help the paper achieve the potential that it seems to carry. The improvements are mainly needed in the directions of the clarity, the quality, and the consistency of the evaluation, the consistent comparison with and detailed discussion of the baselines, and the justification of certain design selections by the authors. Moreover, the value of the proposed key contributions requires further detailed elaboration.
We now procced in a detailed review of the paper. We will use in brackets the lines of the review manuscript to which each comment refers.
Initially, regarding the key contributions of paper:
(80-81) About key contribution 1 “We design a hierarchical localization framework that combines an unsupervised network VAE with traditional pose calculations.”: The added value of selecting VAE as a feature extractor, in comparison to other possible (and potentially existing) approaches, is not clearly shown. Since using the VAE as a feature extractor in this setting is proposed as a key contribution, a natural question that is unanswered is: how does the system performs with and without this contribution? What would be the most similar existing solution and how would it perform?
(82-83) About key contribution 2 “We design the VAE structure for the efficient retrieval of images and train the scheme in order to avoid posterior collapse.”: The paper utilizes State Of The Art (SOTA) methods, which have been proposed for posterior collapse avoidance. Nevertheless, their necessity in this setting has not been demonstrated. Is posterior collapse an occurring problem in this setting? What is the system’s performance with and without the proposed utilization of the SOTA methods for addressing posterior collapse?
(84-85) About key contribution 3 “Our system only requires RGB images as the input and achieves 6DoF pose estimation for largescale indoor scenes. Furthermore, the scheme is easy to implement, efficient and accurate..”: The authors should highlight in a clear way if other unsupervised methods (and more specifically methods using only images) exist and compare against them. Several existing methods are used as baselines in the paper, but their exact nature is not clearly described to the reader. Moreover, the claim that the scheme is 'efficient' (in terms of time) is shown experimentally, though the claim that it is 'accurate' is not in agreement with the experimental results. A more objective characterization of the method’s merits would be desired.
Major remarks:
(2) I would kindly recommend a more descriptive title. Firstly, indoor localization might concern several technologies, and the video-based is not the most often one. Secondly, the overall system has many components, with VAE being mainly its feature extractor, so the "based on a Variational Autoencoder" part of the tittle might not give a spot-on impression about the paper’s content. Given the above, I would imagine something along the lines of: "Hierarchical video-based indoor localization incorporating a VAE-based feature extractor". Nevertheless, it is up to the authors to choose the title of their preference.
(16,74,…) The terms “local and global features” are used repeatedly to describe main concepts of the proposed system. A smooth introduction of these terms and their intuition (probably in the introduction section) would greatly facilitate the reader.
Section 2.1, presenting the related work, needs improvement, in the sense that it should help the reader to clearly understand the similarities and differences of the existing methods in comparison to the proposed method. Which data modalities are needed by each method? Which are unsupervised, as is the proposed method? Maybe a Table would help here, to offer a clear overview, a taxonomy of the existing methods.
(173-174) If no location information of the image is used, I assume that the reconstruction process requires overlapping parts among the images. What are the density requirements of images for this to work? Also, if there is no absolute reference of the images' location information, what is the reference frame of the resulting position estimation? Is it relative to the total size of the 3D model created? Is the model matched somehow to the actual map of the environment? Can we get from the reconstructed 3D space model the notion of distance in meters?
(179) “Covisibility clustering” : Could you please define/explain (and maybe cite) this method?
(188-189) “The VAE encodes the images into the latent space and latent variables are then used as the descriptors”: Which elements of the VAE’s latent space are used as descriptors? Both mean and standard deviation values of the latent space? Only the mean? Or a sample from the distribution?
(188-189) Throughout the paper, there has been no argumentation regarding the necessity of a Variational Autoencoder, in contrast to a simple Autoencoder. Does the stochasticity of the VAE contribute at a certain point or would a simple Autoencoder be a valid option as well? This argumentation would be very helpful.
(224-238) It is mentioned here that KL annealing and free bits methods are used. Earlier in this paper, an extensive presentation of the VAE method took place. The reviewer would expect: 1) a motivation and experimental evidence of why the use of these methods outperforms the basic VAE, in this setting. 2) A more detailed presentation of these methods, equally explicative to the above presentation of the basic VAE, which they apparently outperform. Since overcoming posterior collapse in this setting is presented as one of the three key contributions, more emphasis should be given in this direction, both theoretically (here) and experimentally (later in the paper).
(243) You mention using the COLMAP model to build the cloud model of the scene. Later, in Figure 9, COLMAP appears as a baseline that greatly outperforms the proposed method. Could you please clarify this point? How is COLMAP involved in your method?
(245) I guess this cloud model is built with the database images, offline, much before any online query images appear. Therefore, it would be helpful to the reader if you present this cloud model creation as the first subsection of Section 4, before the VAE presentation. In case that, apart from the raw images, the latent space of VAE is also used in this cloud model creation, this should be explained in more detail.
(267-268) “the camera pose, with the optimal pose selected as the estimation result.” How is a pose selected as optimal? Define optimal
(301-304) “In order to facilitate a comprehensive evaluation, the camera poses of the training set are auxiliary in 3D model building, we use the known poses reconstruction method in COLMAP to establish the models in our experiment, but in theory our system cannot use it and rely on it.” It is not clear to the reviewer: Are the ground truth poses of the training set used for the creation of the 3D cloud model of the building or not? If yes, the claim of proposing an "unsupervised localization method" is compromised. Also, if the system is evaluated with using something that it should normally not used, isn’t the evaluation compromised in the first place?
(245-246) “…only select these images belonging to the Stairs training set as the candidate images.” Doesn't this compromise the goal of dealing with the 7 scenes as one overall deployment? Said otherwise, why retrieve 50 images from the whole training set, if eventually only those from the "correct" dataset are being used?
(361-374) Despite the extensive effort of explanation in the text, the meaning, the intuition and the importance of Figure 7 remains unclear to the reviewer.
(376) Figure 2 reveals an important information about the 7-scenes dataset, that would be very useful to be clarified in the text. From the axes, it appears that the scenes concern very limited areas, of approximatively 2 by 2 meters each. The fact that previous examples throughout the paper (like the image examples and the 3D cloud model) are from a large shopping center, creates to the reader an expectation of an experimental setting of another order of magnitude. It is recommended to describe the spatial size of the 7 scenes earlier on, when the used datasets are introduced.
(377) The fact that 6 other implementations of previous works are used as baseline is impressive! The complete absence of discussion over them in the text (what each method does more or less, if the comparison is consistent or different methods have access to or require the presence of different data, etc.) and an intuitive discussion of what constitutes the advantage of the proposed method, is really missing. Moreover, apart from a table of the median error values, other statistics would be very welcome. Having access to the results, one could easily offer the CDFs of the error, or report quartiles and mean error values. Having implemented 6+1 methods, it is a pity to limit the reported results only to the median error.
(396-408) Some interesting information to add, when discussing the Beidu dataset: What is the size of the deployment area of the Baidu dataset, and specifically the area of ground truth locations (since the building might be dig, but the photos might be taken from a narrow corridor area)? Do the authors claim that the proposed method outperforms the best known performing unsupervised method on this dataset?
(392) Why are 2 different sets of baseline methods are used in Table 1 (evaluating the first dataset) and Figure 9 (evaluating the second dataset)? Also, as suggested regarding the evaluation of the first dataset, more metrics would be welcome in the evaluation of the Baidu dataset as well.
(430) “The system is convenient, easy to implement, and is highly accurate for largescale indoor scenes.” The achieved performance in a large, real life dataset (Beidu dataset), which shows only 24% to 32% of the estimates having an error lower than 5 meters, and with the rest of the error remaining unreported (what's the 50th, 75th, 90th or 95th percentile of error?), hardly justifies the claim that the system "is highly accurate for largescale indoor scenes."
Minor remarks:
Define and introduce acronyms before using them (for instance SfM (44), PnP, RANSAC (270), etc)
English and typo corrections (examples):
- trai[n (171)
- more faster (425)
Terminology, as for example:
- Section 4.2, from line 259 onwards, (as well as Figure 4) discusses a “clustering” that takes place, but then the term “class” is repeatedly used instead of the term “cluster”.
- (197-198) It is mentioned that “At the end of the decoder network, two fully-connected layers are used to output mean”. Probably the word encoder was intended here
(446-447) “Acknowledgments: We are grateful to the reviewers for their suggestions and comments, which significantly improved the quality of this paper.” Kind remark: If this is the first review of the paper, it would be better to add this remark once the reviewer's feedback have been given and the paper quality has been improved :-)
Round 2
Reviewer 2 Report
This is the second review of the manuscript.
The topics discussed by the authors are interesting but the manuscript is still confused and unclear; in addition there are a lot of language errors (e.g. "paraments", "unit Gaussian", "deconder", "with some also needing depth images", "Such as making use", "the preprocessing is offline" just to cite a few) in the changed parts of the manuscript that need to be cleaned.
Some old issues are still present in the current version of the manuscript:
- as far as the reply letter is concerned, please check again the obscure and unclear parts keeping in mind that the issues that have been previously indicated are not language-related only but, mostly, due to oversimplification, missing descriptions, and implicit assumptions thus requiring clarification and discussion. Please check also the following text fragments: (105-106), (142-142), (156), (166-168), (173-175), (202), (208), (213), (306-311), (407-414), (429-431).
- Figure 2 is still unclear and uninformative since it is difficult to map the steps described in the text (e.g., preprocessing, traing, optimization, NN search, 2D-3D matching, etc) into the blocks of the figure. In addition, in the same figure, what is the meaning of the dashed lines? Figure 1 is unclear as well.
- Moreover, the authors failed to include a table to summarized the parameters adopted in their methods. This makes very difficult to understand the method complexity and to evaluate the parameter set needed by the proposed method in practice (e.g. see text at 366-374).
- Some statements such as "precise pose", "achieves 6DoF pose estimation for large scale indoor scenes", "outperforms most deep learned methods", "accurate pose estimation", need some discussions and considerations.
Moreover, there are other general issues:
- which are the advantages of VAE w.r.t. other methods? Please discuss this topic in more details. in addition, the authors say that for VAE "data preparation is minimal": what does it mean and how does it compare with other method requirements?
- It is not clear if the authors have adopted COLMAP for the construction of the 3D model or have used it only for performance evaluation. If COLMAP is employed, is it used off-line or on-line? Please clarify.
- The proposed VAE variants adopted by the authors in Sect. 4.1.3 to solve the posterior collapse problem suffer from some drawbacks. For instance, see the paper "Understanding Posterior Collapse in Generative Latent Variable Models" by James Lucas, George Tucker, Roger Grosse, Mohammad Norouzi, ICLR 2019, and the included references, for details.
- Insert at least a reference in the following parts: (170-171), (296-298).
- VAE methods belong to the family of unsupervised methods. However, do VAE need tagged/labelled RGB images as inputs? Please clarify and discuss.
- Is VAE pre-processing performed off-line or on-line. Moreover, is it scene-dependent or can be applied to different scenarios?
- Figure 8 is unclear: it seems that there are trajectories superimposed on the dots. Please clarify and discuss.
- Please give a name to the proposed methods and clarify the differences between: Ours, Ours Basic, VAE, basic VAE, proposed method, etc.
- Is it possible to produce a table similar to Table 1 also for the results of the Baidu dataset?
- Finally, since indoor localization includes several different families of methods (e.g., radar localization, satellite/pseudolite localization, sensor-based localization, radio active/passive localization, video localization), please consider revising the title.
Reviewer 3 Report
The authors have made a significant effort to intergrade the reviewer's remarks and to improve the paper.
The paper proposes a system whose performance is evaluated, and its results are compared to the reported performance of other published works.
Nevertheless, even if as a proof of concept it has its merits, I am not convinced on the practical significance of the proposed solution in the real world. The method claims to be unsupervised, in the sense that it does not need access to images whose location is known. The localization method indeed does not require the direct use of images marked with a ground truth location for its training. Nevertheless, it indirectly does, since it requires a 3D model of the space, which is actually built by using a dataset of images marked with their ground truth location. Moreover, the COLMAP method is used by the authors to build this model. COLMAP is later used as a baseline against which the proposed method is compared to, and COLMAP greatly outperforms the proposed method.
The questions that arise are the following:
- A) If the collection of a dataset of images marked with their ground truth location is needed by the proposed method in order to create the 3D model that the proposed method uses, why not use this collected information for the localization phase?
- B) If we already went into the trouble of having collected a dataset of images marked with their ground truth location, and we used COLMAP for building the 3D model, why not use COLMAP for localization as well, since it is more accurate?
Some other points:
1) The authors mention in their response #10 that ("...encodes the semantic 3D voxel volumes into latent variables, which are chosen as global descriptors.."), but in fact in the previous responce #9 they mentioned only using the mean, thus a single point, and not sampling, which would indicate actually using the volume. Intuitively, the key about VAE's stochasticity is exactly the use of distributions (based on mean and st.d.) from which we may sample. The fact that different embeddings might occur in different trainings can happen in both VAE and normal AE according to the weight initialization and the random seed. Could the authors argument at which point in their system the stochasticity of a VAE is an advantage over a normal AE, given the fact that they simply use the mean value and do not sample from the distribution?
2) Please, clearly highlight if the reported results of baselines are not from re-implementing these methods, but a direct transfer of their reported results on their papers (as for instance in your responses 12 and 21). Also, please mention if all results (of the baselines and of your proposed method) are reported on the same test set.
3) Do the authors claim that the proposed method outperforms the best known performing unsupervised method on this dataset?
